# Appropriate Irrigation and Fertilization Regime Restrain Indigenous Soil Key Ammonia-Oxidizing Archaeal and Bacterial Consortia to Mitigate Greenhouse Gas Emissions

Liang Xiao [1], Libin Bao [2], Lantian Ren [2], Yiqin Xie [2], Hong Wang [2], Xiang Wang [2], Jianfei Wang [2], Cece Qiao [1,2,*] and Xin Xiao [2,*]





[1] Jiangsu Provincial Key Lab of Solid Organic Waste Utilization, Jiangsu Collaborative Innovation Center of Solid Organic Wastes, Educational Ministry Engineering Center of Resource-Saving Fertilizers, Nanjing Agricultural University, Nanjing 210095, China; 2016203037@njau.edu.cn

[2] Department of Environmental Science and Engineering, Anhui Science and Technology University, Donghua Road 9#, Chuzhou 233100, China; baolibin@ahstu.edu.cn (L.B.); renlt@ahstu.edu.cn (L.R.); xieyq@ahstu.edu.cn (Y.X.); wanghong@ahstu.edu.cn (H.W.); wangxiang@ahstu.edu.cn (X.W.); wangjf@ahstu.edu.cn (J.W.)

\* Correspondence: qiaocece@ahstu.edu.cn (C.Q.); xiaoxin@ahstu.edu.cn (X.X.); Tel.: +86-0550-6732656 (X.X.)

**Abstract:** Harnessing an ammonia-oxidizing microbiome has become an increasingly attractive form of management for mitigating greenhouse gas emissions in rice paddies; however, the relationship between greenhouse gas emissions and ammonia-oxidizing microbiomes, using a nitrogen application and irrigation regime, has not been well investigated. To decipher which of (and how) the specific mmonia-oxidizing bacterial species drive the greenhouse gas $CH_4$ and $N_2O$ emissions, a field experiment with varying nitrogen application and irrigation regimes was initiated to investigate the succession of key bacterial consortia associated with GHG emissions. The results showed that water-saving irrigation (AWD) significantly increased $NO_3$-N and $NH_4^+$-N concentrations, compared with conventional irrigation (FDF), whereas (total nitrogen) TN was little higher in FDF (1.38 g kg$^{-1}$) compared with the AWD (1.36 g kg$^{-1}$). During the rice-growing season, $CH_4$ emissions ascended speedily, and emissions peaked at maximum values of 3.32 and 4.41 ug mg$^{-2}$ h$^{-1}$ on day 5 in FDF and AWD irrigation regimes, respectively, and then they rapidly decreased during the midseason period, maintaining a relatively low emission rate until the rice was harvested. The patterns of $N_2O$ emission fluxes had the same tendencies with N fertilization. Putative key taxa, such as *Flavobacterium*, *Massilia*, *Arenimonas*, *Novosphingobium*, *Pseudomonas*, exhibited significant positive relationships with higher GHG emissions, suggesting that they make particularly obvious contributions to $N_2O$ emissions. These putative taxa should be considered when designing a high nitrogen application and irrigation strategy. As such, the nitrogen application of N180, and the irrigation regimes of water-saving irrigation, are recommended methods for N conservation and the mitigation of greenhouse gas emissions in rice paddies.

**Keywords:** irrigation and fertilization regimes; ammonia-oxidizing microbiome; greenhouse gas emissions

## 1. Introduction

It has been the consensus that GHG emissions from agricultural soils are of principal relevance for global warming [1]. Simultaneously, water resources have increasingly become a serious problem due to imbalanced precipitation and greater demands for crop production globally [2]. To address these problems, practices such as the intermittent irrigation of alternating wetting and drying, coupled with appropriate fertilization, have become increasingly attractive methods for sustainable agriculture management [3].

As a main agricultural crop, rice is cultivated in tropical and subtropical regions, which cover about 20% of global farming land [4]. Remarkably, waterlogged rice paddies

have been confirmed as a main source of the anthropogenic GHG emissions, comprising approximately 10% $N_2O$ and 50% $CH_4$ of the emissions of agricultural soils, respectively [5].

Water management has been identified as a determinant regulating $N_2O$ and $CH_4$ emissions in rice paddies. For instance, Berger et al. [6] reported continuous flood irrigation, resulting in a greater emission of both $N_2O$ and $CH_4$ by 200% and 70%, respectively, compared with intermittent and minor flood irrigation managements. Various irrigation practices in rice paddies are managed in China [7].

It cannot be ignored that, in previous decades, a large amount of nitrogen-based (N) fertilizer greatly contributed to high rice grain yields [8]; however, there is also an increasing concern that the excessive use of N fertilizer has greatly contributed to the $N_2O$ concentrations, resulting in GHG emissions [9]. As a main agricultural practice, the N input promotes the nitrification and denitrification process, and thus was recognized as regulating GHG emissions from the soil [10]. $N_2O$ emissions exponentially increased when the N inputs exceeded the reasonable rate [11], whereas the optimum N application presented a contrary result [12].

To mitigate the steady increase in GHG loading, it is necessary to better understand the underlying microbial mechanisms leading to soil GHG formation. Accumulating evidence indicates the central role of soil microbiome, especially ammonia-oxidizing archaea (AOA) and bacteria (AOB), in regulating the major processes of carbon/nitrogen transformations and GHG emissions [13]. The abundance, composition, structure, and physiology patterns of GHG-producing and -consuming organisms have become a concern in rice paddies [14]. For instance, the soil AOA and AOB communities have been observed as both taxonomic and functional successions that are influenced by water management [15] or fertilization regimes [16]; nevertheless, they are needed to identify factors that affect the soil AOA and AOB communities to minimize the soil environmental impacts. However, although most studies to date have addressed either the estimation and simulation of agricultural GHG fluxes [17] or the impact of environmental factors on GHG fluxes [18], relatively little research has been directed toward parameterizing and integrating microbial pathways into rates of GHG fluxes in soil ecosystems. Consequently, soil microbial ecology still remains a challenge in terms of enabling practical mitigation methods for GHG emissions. Accurate identification of these critical microbially mediated GHG regulating processes might be a prerequisite for developing the next generation of microbially oriented ecosystem models. There is still a lack of effort concerning the specific interactions between water dynamics and N fertilizer applications on the impact of GHG emissions and nutrient availability in the field environment. These interactions could effect the AOA and AOB communities through the soil microbial community, and thus, they can be either beneficial or detrimental to GHG emissions; therefore, identifying the major microbial reason concerning GHG production from soils will be critical for mitigation strategies in agricultural management.

Overall, understanding variations in the AOA and AOB community compositions and functions is useful for predicting the GHG regulating processes. Monitoring water management and N fertilization, coupled with an understanding of AOA and AOB communities' variation in paddies, will provide further insight into the GHG regulating processes. As such, this study aims to: (1) decipher how water management, N fertilization, and their interactive effects influence the succession of the soil in AOA and AOB communities; (2) investigate the possible mechanisms responsible for GHG regulating processes; and (3) identify key microbial species responsible for GHG emissions.

## 2. Materials and Methods

### 2.1. Study Site and Experimental Design

The field experiment was performed in a plant science park (32°86′ N, 117°4′ E, 5.2 m a.s.l), located in Anhui Province, one of the major areas of rice and wheat production in China (Figure 1). This region has a subtropical monsoon climate, with a mean annual precipitation of 1236.0 mm. Moreover, the period of April to June contributed to almost 50% of the rainfall. The soil in this experimental field was characterized as hydromorphic,

with a pH of 7.11 (5:1 water:soil, *v/w*), containing 68.70 mg kg$^{-1}$ available N, 12.81 g kg$^{-1}$ dissolved organic carbon, 33.58 mg kg$^{-1}$ available phosphorus, and 51.63 mg kg$^{-1}$ available potassium in the surface soil layer (0–20 cm).

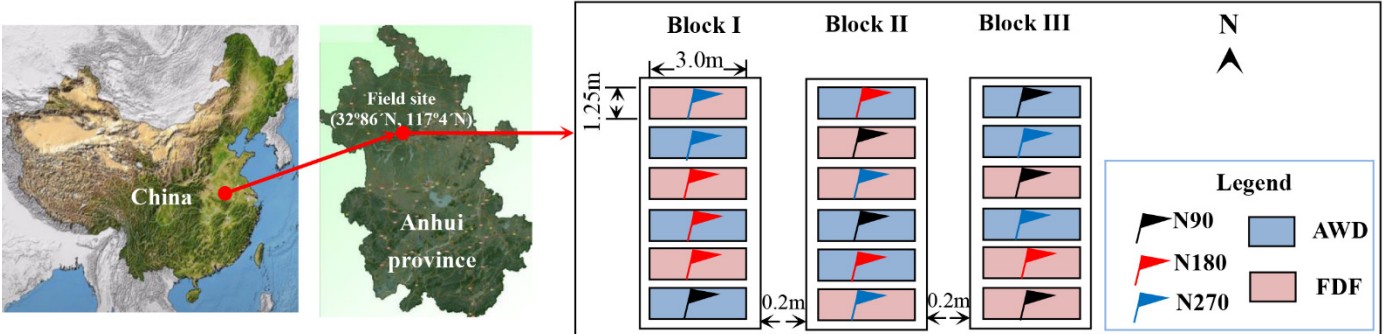

**Figure 1.** The schematic diagram of soil sampling sites in Anhui province, China. AWD: alternating wetting and drying; FDF: flooding-midseason drainage-flooding.

A completely randomized block design experiment with an irrigation regime and N application rate was conducted. To be precise, the irrigation regime includes: (1) water-saving irrigation, namely, alternating between wetting and drying (AWD), and (2) conventional irrigation, named the flooding-midseason drainage-flooding (FDF) regime. The FDF plots were flooded (30–50 mm of water depth) 3 days before rice transplanting on 8 June until 11 July 2019, followed by a midseason drainage from 11 July to 21 July, and then they were re-flooded until 7 days before harvesting. For the AWD treatments, flooding was initiated until plants were revived on 15 June, which were followed by alternating wetting and drying cycles until 7 days before harvesting. Three N application regimes were also established: (1) 90 kg N ha$^{-1}$ (N90); (2) 180 kg N ha$^{-1}$ (N180); and (3) typical of fertilized areas in this region, 270 kg N ha$^{-1}$ (N270). Taken together, six treatments were established with three replicates for each of the treatments. Each plot was 3.75 m$^2$ (3 m × 1.25 m) and isolated by brick concrete (1.2 m depth) between plots. All the plots were arranged under a rain shelter to control the irrigation water usage. "Gangyou 527", a typical rice cultivar in this region, was sown in a nursery bed on 29 April, and then rice seedlings were transplanted on 8 June and harvested on 8 October 2019. Half of the urea as N fertilizers were applied as basic fertilizers on 4 June, 20% was applied at the tillering stage on 5 July, and 30% was applied at the heading stage on 2 September. Calcium superphosphate (containing 12% $P_2O_5$) as the phosphorous fertilizer employed 75 kg ha$^{-1}$, and potassium sulfate (containing 60% $K_2O$) as the potassium fertilizer employed 150 kg ha$^{-1}$ in all treatments as the basal fertilizer.

### 2.2. $N_2O$ and $CH_4$ Fluxes Measurements

Fluxes of $N_2O$ and $CH_4$ were simultaneously measured using a static vented chamber-based method [19]. In brief, PVC flux chambers (50 cm width × 50 cm length × 50 cm depth) were installed in each replicate plot before rice transplanting for gas sampling, as well as top sampling chambers that varied from 50 cm to 100 cm in height depending on plant height according to Wang et al. [7]. Gas fluxes were determined three times per week between 9:00 a.m. and 11:00 a.m. over the rice growth period. The top chambers were mounted on the base frame, gas samples were then collected with a 60 mL syringe at 5 min intervals during the closure time of 15 min (i.e., at time 0 and after 5, 10, and 15 min), and then gas samples were stored in evacuated vials for <24 h before analysis.

$N_2O$ and $CH_4$ emissions in the samples were simultaneously tested on a gas chromatograph (Agilent 7890A, Gow Mac Instrument Company, Bethlehem, PA, USA) equipped with an electron capture detector (ECD) at 330 °C [20]. Gas fluxes were calculated from (non)linear regressions of consecutive samples, such as GHG concentration against the chamber closure time. Seasonal cumulative $N_2O$ and $CH_4$ emissions were subsequently calculated according to Maucieri et al. [21].

### 2.3. Soil Sampling, Properties Analysis and DNA Extraction

Soil samples were collected during the rice growth period (based on the dynamics of $N_2O$ emissions). Four sub-samples were randomly collected and mixed as a composite soil sample from each plot. Each sample was divided into three parts, one portion of each sample was stored at 4 °C, one was stored at −80 °C for subsequent DNA extraction, and the other was air-dried for the determination of physicochemical properties including pH, total organic carbon (TOC), $NH_4^+$-N, $NO_3^-$-N, available P (AP), available K (AK), total N (TN), total P (TP), and total K (TK), according to Qiao et al. [22].

### 2.4. DNA Extraction and Sequencing

Total genomic DNA was extracted from 0.25 g soil subsamples. The sequencing of gene amplicons and the subsequent data analysis were performed using a MiSeq DNA sequencer (Illumina, San Diego, CA, USA) by Illumina Miseq sequencing at Biomarker Biotechnology Co., Ltd, Beijing, China.

### 2.5. Sequence Data Processing

After the removal of singleton OTUs, a total of 314,208 AOA and 407,448 AOB gene sequences remained for microbial community analyses, respectively. The number of sequences per sample harbored an average of 17,456 AOA and 22,636 AOB sequences, respectively.

The classification of OTU was performed using the RDP classifier.

### 2.6. Statistical Analysis

All statistical tests were determined at the 0.05 probability level. MANOVA was used to determine the effects of irrigation management and N application, as well as their interactions on the soil's physicochemical parameters.

Microbial β-diversity patterns across nitrogen application and irrigation regime treatments was assessed with Non-metric multidimensional scaling (NMDS). ANOSIM and PERMANOVA were conducted to investigate community (dis)similarities using the vegan package of the R (version 4.0.4, R Core Team., 2020). In addition, Pearson correlation coefficients between the abundances of putative key bacterial consortia such as *Flavobacterium*, *Massilia*, *Arenimonas*, *Novosphingobium*, *Pseudomonas*, and GHG emissions were also calculated. If the multivariate F value was significant, then the individual univariate analysis would proceed.

## 3. Results and Discussion

### 3.1. Variations of Yield

The effects of the irrigation strategy and nitrogen application on rice yield were shown in Figure 2. Rational irrigation regimes and nitrogen management are important factors affecting rice yield. Nitrogen applications could effectively enhance rice tillering, but would lead to an increased ineffective division, and would even be prone to lodging and having a low seed setting rate [23]. Although the irrigation regime had no significant effect on rice grain yield, AWD irrigation enhanced the grain yield slightly by 7% (Figure 2), which is consistent with previous research by Chu et al. [24]. Generally, the results of this study show that an increase in rice grain yield is associated with an increase in N fertilization; this might be attributed to N fertilized enhanced rice grain numbers per spike (Table 1), which is consistent with previous research by Wang et al. [25]. The theoretical yield of rice under a nitrogen application of N180, and irrigation regimes of water-saving irrigation treatment, was the highest, which, in accordance with previous studies, indicates that synergistic water–N interaction should be adopted for a high grain yield [25].

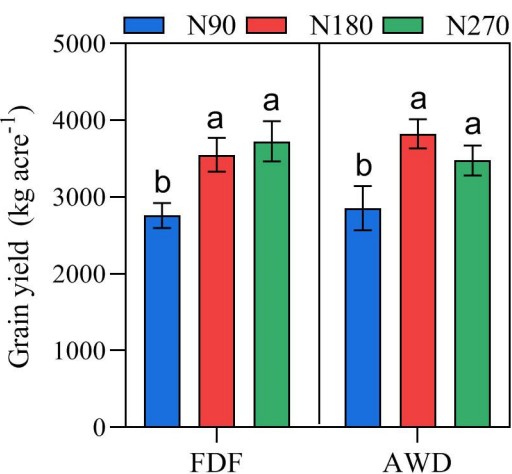

**Figure 2.** Rice grain yield as affected by the interaction of a N application with different irrigation regimes. The bars indicate the standard error of the means (±SE). Different letters indicate significant differences according to the Fisher's LSD test.

**Table 1.** Effect of optimized fertilization on yield components of rice.

| Items | Panicles ($\times 10^4 \cdot ha^{-1}$) | Grains per Panicle | Filled Grains (%) | 1000-Grain Weight (g) | Yield (kg.acre$^{-1}$) |
|---|---|---|---|---|---|
| | | Nitrogen application (NA) | | | |
| N90 | 12.17 ± 1.89 a | 176.09 ± 3.29 b | 91.76 ± 1.20 a | 28.40 ± 0.66 a | 2808.04 ± 68.59 b |
| N180 | 11.91 ± 0.15 a | 202.27 ± 5.75 a | 94.03 ± 1.57 a | 28.53 ± 0.81 a | 3688.37 ± 191.75 a |
| N270 | 12.59 ± 1.29 a | 197.67 ± 4.51 a | 94.30 ± 1.77 a | 28.35 ± 1.24 a | 3603.03 ± 177.18 a |
| | | Irrigation regime (IR) | | | |
| FDF | 11.16 ± 1.00 a | 191.65 ± 13.73 a | 93.09 ± 3.59 a | 28.70 ± 0.67 a | 3346.87 ± 516.19 a |
| AWD | 12.11 ± 1.34 a | 192.37 ± 15.13 a | 93.63 ± 5.36 a | 28.18 ± 0.82 a | 3386.08 ± 490.21 a |
| | | ANOVA *p*-values | | | |
| NA | NS | NS | NS | NS | <0.01 |
| IR | NS | NS | NS | NS | NS |
| NA*IR | NS | <0.05 | NS | NS | <0.05 |

NA, effect of nitrogen fertilizer on rice yield and yield components; IR, effects of irrigation patterns on yield and yield components of rice; NA*IR, effects of the interaction between water and nitrogen on yield and yield components of rice; NS means not significant. Means followed by the same letter for a given factor are not significantly different ($p > 0.05$).

### 3.2. Variations of Soil Physicochemical Parameters

Typically, both of the N180 and N270 treatments resulted in significantly higher levels of soil $NO_3^-$-N, $NH_4^+$-N and total nitrogen (TN), and a decrease in pH as well as total organic carbon (TOC), compared with N90 (Table 2). The significant physiochemical difference between the N180 and N270 amendments was also observed with $NO_3^-$-N, $NH_4^+$-N, and TN. Accordingly, the concentrations of $NO_3^-$-N, $NH_4^+$-N were significantly affected by the irrigation regime, whereas only $NH_4^+$-N was significantly affected by the interactions between the nitrogen application and irrigation regime. Both $NO_3$-N and $NH_4^+$-N exhibited a clear increase in water-saving irrigation (AWD) compared with conventional irrigation (FDF), whereas, conversely, TN was little higher in FDF (1.38 g kg$^{-1}$) compared with the AWD (1.36 g kg$^{-1}$). This conclusion is consistent with the findings of Yang et al. [26]. In particular, no significant differences in soil TOC were observed between the nitrogen application and the irrigation regime, or in their interactions. From the perspective of economic and soil physicochemical properties, N180 and water-saving irrigation constitutes an effective management strategy for sustainable agriculture.

**Table 2.** Effects of the nitrogen application, the irrigation regime, and their interactions on selected physiochemical characteristics of paddy soils.

| Items | pH | TOC (g kg$^{-1}$) | NH$_4^+$-N (mg kg$^{-1}$) | NO$_3^-$-N (mg kg$^{-1}$) | Total N (g kg$^{-1}$) |
|---|---|---|---|---|---|
| | | Nitrogen application (NA) | | | |
| N90 | 7.17 ± 0.04 a | 13.79 ± 1.54 a | 69.50 ± 4.61 c | 3.81 ± 0.14 b | 1.18 ± 0.08 c |
| N180 | 7.04 ± 0.09 b | 13.44 ± 1.71 a | 75.76 ± 2.64 b | 4.21 ± 0.54 b | 1.35 ± 0.06 b |
| N270 | 6.96 ± 0.05 b | 13.46 ± 0.87 a | 86.80 ± 3.78 a | 5.30 ± 0.54 a | 1.59 ± 0.13 a |
| | | Irrigation regime (IR) | | | |
| FDF | 7.03 ± 0.10 a | 13.01 ± 1.16 a | 75.45 ± 8.7 b | 4.24 ± 0.67 b | 1.38 ± 0.02 a |
| AWD | 7.08 ± 0.11 a | 14.12 ± 1.35 a | 79.26 ± 7.54 a | 4.64 ± 0.86 a | 1.36 ± 0.20 a |
| | | ANOVA *p*-values | | | |
| NA | <0.01 | NS | NS | <0.01 | <0.001 |
| IR | NS | NS | <0.05 | <0.05 | NS |
| NA*IR | NS | NS | <0.05 | NS | NS |

Values are means with standard deviations (n = 8 or n = 12). NS: not significant. Means followed by the same letter for a given factor are not significantly different ($p > 0.05$).

### 3.3. Variations of GHG Emissions

During the rice-growing season, similar temporal trends in CH$_4$ fluxes were observed for both irrigation regimes (Figure 3B). CH$_4$ emissions ascended speedily, and emission peaks at maximum values of 3.32 and 4.41 ug mg$^{-2}$ h$^{-1}$ on day 5 in FDF and AWD irrigation regimes, respectively, occurred approximately one week after rice transplanting at the waterlogged stage. Among which, the N270 harbored the highest value. Thereafter, CH$_4$ emissions rapidly decreased during the midseason period, and then they were maintained at a relatively low emission rate until the rice was harvested. This variation is supported by previous studies which addressed the fact that the AWD regime significantly decreased the GWP compared with FDF [7].

The patterns of N$_2$O emission fluxes harbored the same tendency with N fertilization, which followed different variations with the irrigation regime in rice paddies (Figure 3C,D). The majority of N$_2$O emissions occurred in the midseason drainage, and the second top-dressing also produced a weak peak flux of N$_2$O in all the plots. To be precise, N$_2$O emissions of the non-waterlogged period accounted for 60–64% of the seasonal total. Obvious single N$_2$O flux peaks occurred for FDF due to midseason drainage, and thereafter, it decreased sharply until imperceivable N$_2$O fluxes with a re-flooding extension throughout the whole rice-growing season occurred (Figure 3A). N$_2$O was mainly emitted during the 15 day drainage period. Under the AWD irrigation regime, however, N fertilization harbored the flux peaks of N$_2$O during the moist period (Figure 3B). This lower GHG emission supports higher efficiency as an effective management tool. To conclude, water-saving irrigation would be expected to occur in a similar way to relatively low GHG emissions, with the aim of achieving similar results.

### 3.4. Succession of Ammonia-Oxidizing Microbiomes

Non-metric multidimensional scaling (NMDS) was implemented to investigate how (dis)similar the AOA and AOB communities' compositions were in each treatment (Figure 4). The treatments indicated significantly different AOA communities within both the irrigation management (PERMANOVA, pseudo-F = 4.03, $p < 0.001$; ANOSIM, $p < 0.001$) and N application of the AOA community. Similarly, the AOB community was also statistically significant. The highest explanatory factor was the irrigation regime that contributed 12.90% ($p = 0.010$) and 17.38% ($p = 0.061$) of the total AOA community variation, respectively, whereas 8.89% ($p = 0.022$) and 9.73% ($p = 0.001$) of the AOB community total variation was attributed to nitrogen application. Interestingly, the AOA community's variational tendency was similar to the AOB community in terms of its irrigation regime and nitrogen

application. This may suggest that the AOA and AOB community compositions were impacted through similar mechanisms. This impact upon AOA and AOB community compositions has been previously found to be due to changes in the field environment, such as soil aggregate size [9], N management strategy [27], biochar application [28], and irrigation regime [26]. Thus, adjustment of the nitrogen application and irrigation regime would be expected to significantly impact AOA and AOB community compositions.

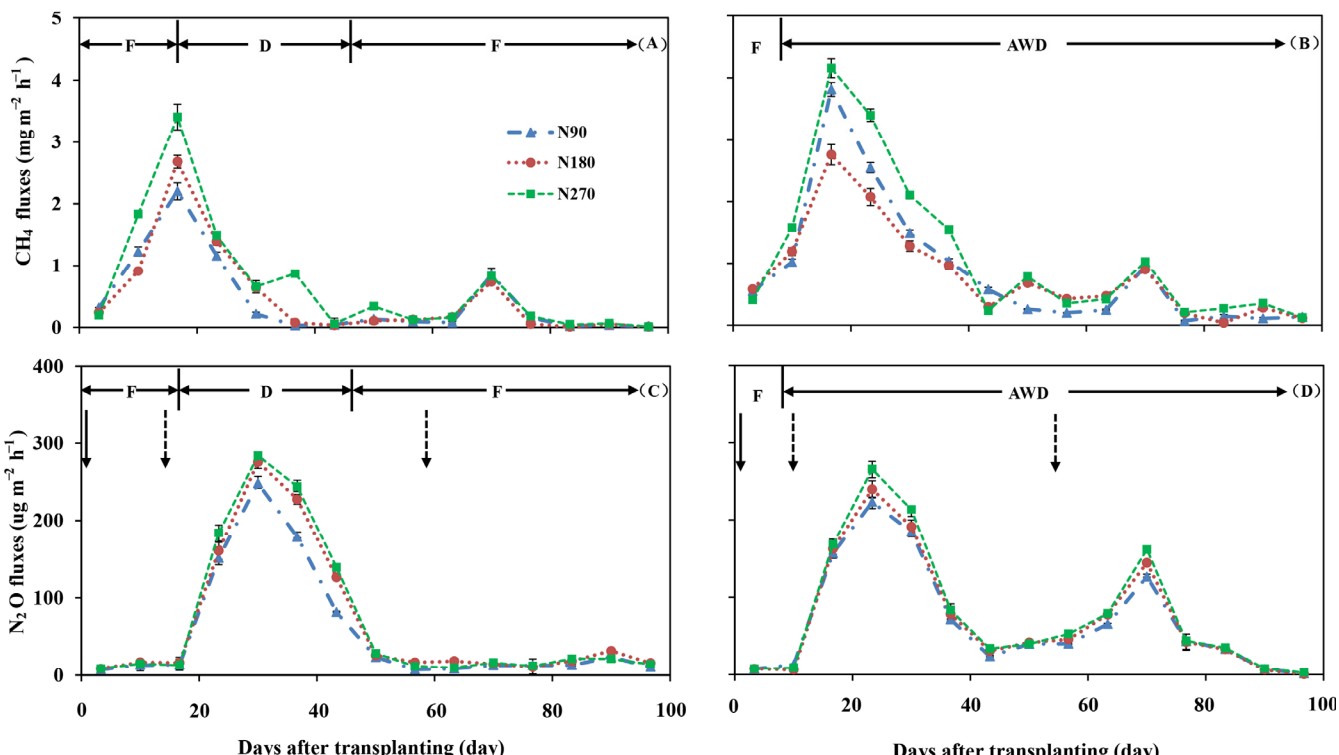

**Figure 3.** Seasonal dynamics of $N_2O$ (**A**,**B**) and $CH_4$ (**C**,**D**) emission fluxes from conventional water irrigation (FDF) and water-saving irrigation (AWD) management in rice paddies. F, D, and AWD represent flooding, midseason drainage, and moistness by alternating wetting and drying cycles, respectively. Solid and dotted arrows represent the basic and top-dressing application of N fertilizer. The legend in Figure (**A**) applied to the other.

### 3.5. The Key Taxa Potentially Linked to GHG Emissions

To identify putative functionally specialized AOA and AOB groups, Spearman's correlation coefficients were calculated between the average $N_2O$ emissions and relative abundances of the enriched genera (Figure 5). Only genera that were significantly correlated with GHG emissions were analyzed. Relative abundances of *Flavobacterium* (r = 0.665; $p < 0.05$), *Massilia* (r = 0.576; $p < 0.01$), *Arenimonas* (r = 0.496; $p < 0.05$), *Novosphingobium* (r = 0.664; $p < 0.01$), and *Pseudomonas* (r = 0.665; $p < 0.05$) exhibited significantly positive relationships with average $N_2O$ emissions. In contrast, *Sporacetigenium* (r = −0.472; $p < 0.05$) was negetively correlated. Taken together, our data suggests that enrichments of *Flavobacterium*, *Massilia*, *Arenimonas*, *Novosphingobium*, and *Pseudomonas* were most associated with the measured GHG emissions, and thus played vital roles in the GHG emossion process.

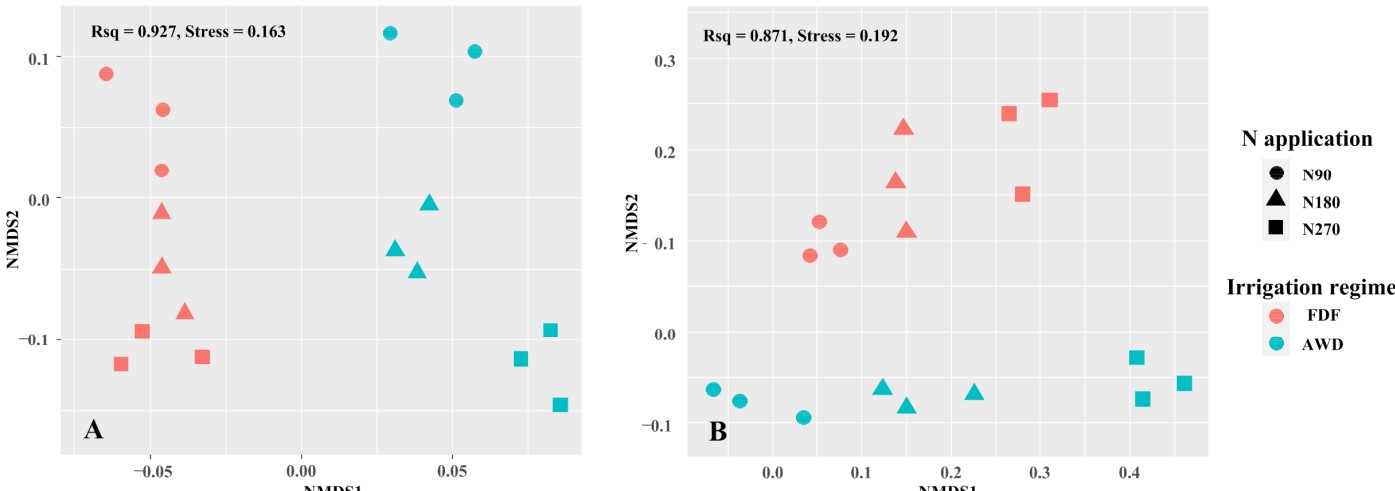

**Figure 4.** Non-metric multidimensional scaling (NMDS) ordinations of AOA (**A**) and AOB (**B**) communities. Samples from different fertilizer treatments both in maize (square) and cabbage (circle) are marked by different colors.

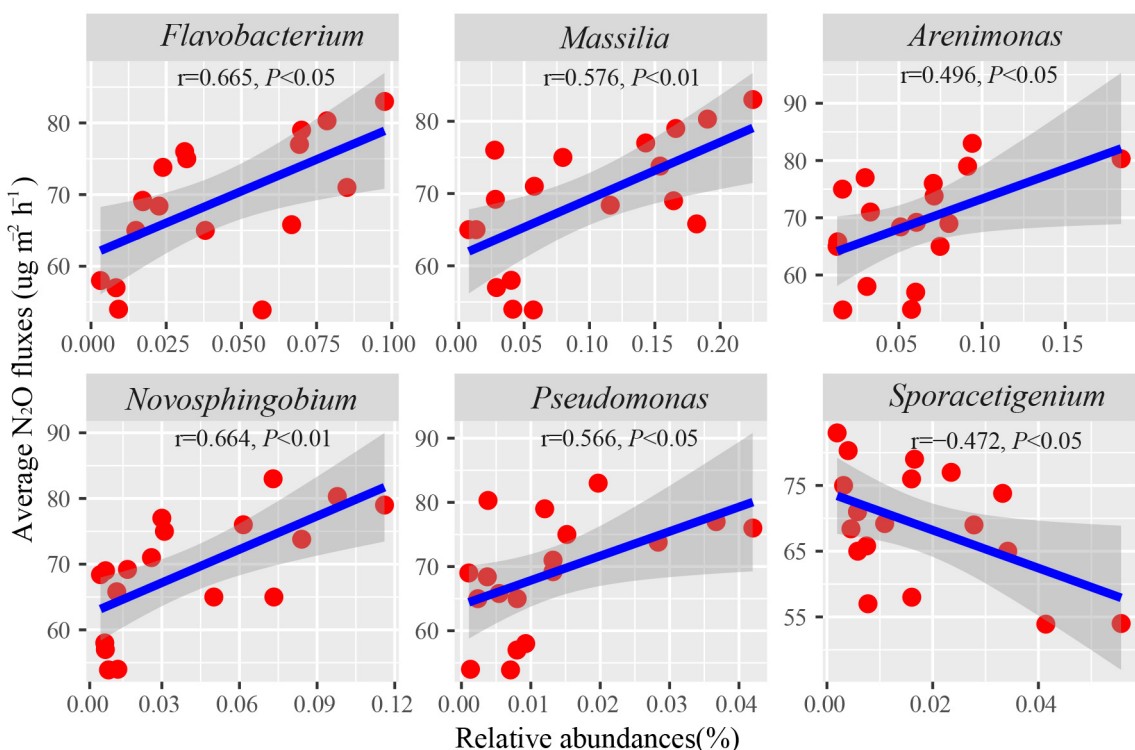

**Figure 5.** Spearman's rank correlation (r) between the relative abundances of candidate genera and the average GHG emissions. All of the taxa were subjected to the Spearman correlation analysis. Taxa with *p* > 0.05 are not shown.

Among those enriched genera, the genera *Flavobacterium* and *Pseudomonas* have been reported to play key roles in N cycling, especially in the denitrifying process [29]. Simultaneously, *Novosphingobium* was documented as a metabolically versatile member, playing a significant role in the N cycle [30]; *Massilia* has been identified as being related to the N metabolism, and it was detected as being a putative keystone denitrifier [31]. *Arenimonas* was capable of removing nitrate, and is used for stable-isotope probing techniques [32]. In the present study, higher relative abundances of the above genera were observed in

N270 treatments; however, for the sake of practical investigation, there are few reports concerning greenhouse gas (GHG) emissions of the rest of the genera.

SEM (Figure 6) further indicated that the identified key AOA and AOB species were influenced by suitable nitrogen application and irrigation regimes, which resulted in altering the contents of $NH_4^+$ and TN, thus appearing to significantly contribute to GHG emissions.

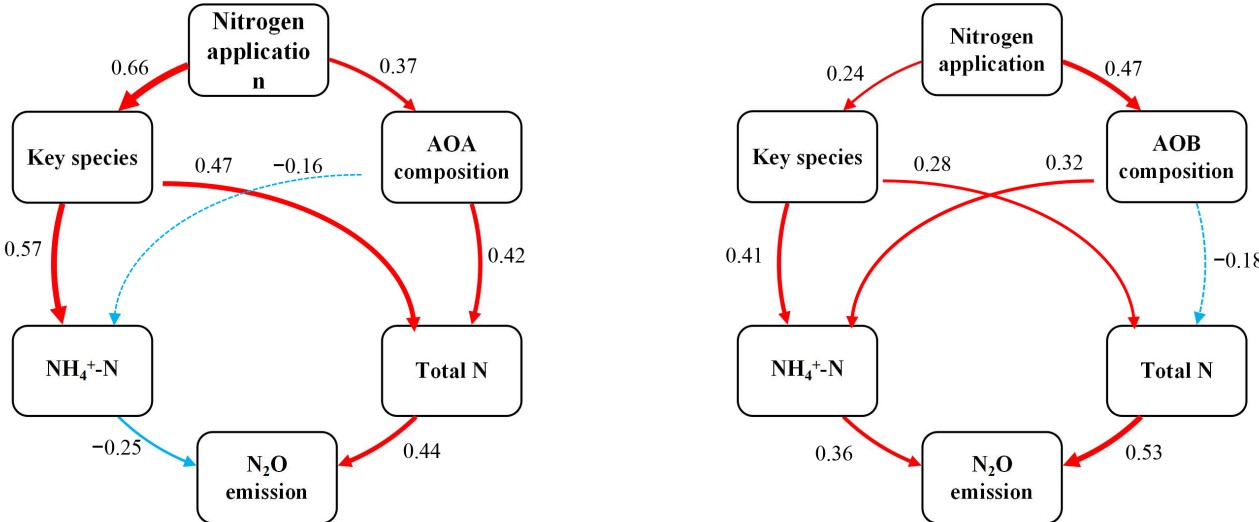

**Figure 6.** Path diagrams obtained from structural equation modeling illustrating the effects of nitrogen applications and irrigation regimes on GHG emissions as a result of changes in AOA and AOB communities' composition and the relative abundances of key taxa.

## 4. Conclusions

In summary, this study highlights a suitable nitrogen application and irrigation regime that induces low GHG emissions, and its influence is proposed as being the result of mechanisms such as: (1) the impacting ring in archaeal and bacterial community compositions; (2) selected key taxa such as *Flavobacterium*, *Massilia*, *Arenimonas*, *Novosphingobium*, and *Pseudomonas*; and (3) expressions and activities that are potentially linked to key taxa. As a result, N180 and water-saving irrigation holds promise as a sustainable strategy for the mitigation of greenhouse gas emissions.

**Author Contributions:** C.Q., L.X., L.B. and X.X. conceived and designed the experiments; L.R., Y.X. and H.W. performed the experiments; C.Q., X.W. and J.W. analyzed the data. L.X. and C.Q. wrote the paper. All authors reviewed and edited the manuscript. All authors have read and agreed to the published version of the manuscript.

**Funding:** This research was supported by Major Science and Technology Projects in Anhui Province (18030701214), the Natural Science Foundation of the Anhui Higher Education Institutions of China (KJ2020A0050), Anhui Provincial Natural Science Foundation of China (2108085QAWD26, 2008085QD181), and the Anhui Province Major Science and Technology Project (201903a06020023), 'Technical specification for precision fertilization of selenium enriched rice' (2021gnny03).

**Institutional Review Board Statement:** Not applicable.

**Informed Consent Statement:** Not applicable.

**Data Availability Statement:** The raw sequence data reported in this paper have been deposited in the Genome Sequence Archive of the BIG Data Center, Chinese Academy of Sciences, under accession code CRA006798.

**Conflicts of Interest:** The authors declare no conflict of interest.

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
