# Peer review of "Appropriate Irrigation and Fertilization Regime Restrain Indigenous Soil Key Ammonia-Oxidizing Archaeal and Bacterial Consortia to Mitigate Greenhouse Gas Emissions"

_sustainability, doi:10.3390/su14106113_

Round 1

Reviewer 1 Report

I want to congratulate the team for the research conducted and how the article was written. The article is clear and concise, presenting the main scientific information.

My remarks are editorial and not scientific:

  1. in table 1 please arrange column no. 3, more exactly TOC (g kg-1) - to be like the other columns
  2. also in table 1 - we have a series of data, but we also have the letters a, b and c. wath their meaning is not found in the text related to the table, nor as notes of the table.
  3. on line 221 we have a notation FIG. 1B, and on line 236 FIG. 1A. but in figure 1 we do not have the notations A and B. can reference be made to figure 2 ??? 
  4. on line 249 you have Fig. 2 - I think it should be Fig. 3.
  5. on line 303 - Fig. 5

Author Response

Dear Editor:

Thank you very much for your kind assistance in our above manuscript. We appreciate all the time and effort you put in the handling of our manuscript. We have modified the format according to your request. Our description on revision according to the editors’ comments was attached as the main document. A revised manuscript with the correction sections red marked was attached as the main document and for easy check/editing purpose.

Thank you for the kind advice.

We are looking forward to your further comments.

Sincerely yours,

Liang Xiao

Reviewer 2 Report

Thank you for the opportunity to read this MS. I hope this feedback will find you with great spirit to improve the MS. The authors have an interesting experiment and showed information about the “Appropriate irrigation and fertilization regime restrain indigenous soil key ammonia-oxidizing archaeal and bacterial consortia to mitigate greenhouse gas emissions”.

Comments:

Abstract:

line 18 clarify the abbreviation (TN)

Line 23, 24: The authors mentioned putative key taxa, such as Flavobacterium, Massilia, Arenimonas, Novosphingobium, Pseudomonas, materials and methods of MS did not contain any information about them.

Line 24: Replacing Pseudomonas with Pseudomonase

Keywords: 

  • Irrigation; fertilization regimes (correct).
  • Remove Methan and Nitrous oxide.
  • Add greenhouse gas emissions

Introduction:

Introduction can be shortened.

Authors mentioned it can’t be ignored that large amount of nitrogen-based (N) fertilizer greatly contribute to high rice grain yields in the past decades, so why did the authors ignore the yield estimation in this MS?

Materials and Methods:

What is the volume of water used in irrigation for each treatment? please clarify that

Figure 1 add description of AWD and FDF

Authors mentioned each plot was 3.75 m2, i see it is very small.

Line 127, 128: for the AWD treatments, flooding was initiated until plant reviving in 15 June, and then followed by an alternating wetting and drying cycles until 7 days before harvesting. How to perform wetting and drying cycles ?

Line 161: Authors mentioned half of the urea as fertilizer N fertilizers were applied as basic fertilizers on 4 June and 20% at tillering stage on 5 July and 30% at heading stage on 2 September, What about the remaining amount of nitrogen? and what is the method of adding fertilizer?

What are the dates for adding different fertilizers?

Results and Discussion:

line 205 Typically  instead of Tipically

Table 1 Effects of nitrogen application and irrigation regime and their interactions on selected physiochemical characteristics for paddy soils, Where are the overlap values between the treatments?

Why is there no Table for yield?

Conclusions:

N180 and water-saving irrigation holds promise as a sustainable strategy for the greenhouse gas emissions mitigation; on what basis do we recommend it without linking it to the yield?

References:

The references paragraph must be rewritten on the MDPI format.

Author Response

(The authors gave the same response as above.)

Reviewer 3 Report

1: "Type of the Paper (Article, Review, Communication, etc.) - for change.

14: "CH4 and N2O" - Please use subscript throughout the paper.

18: "g kg-1" - Missing superscript and point.

107: I suggest that to specify the aim of the work

218 Please correctly write down the title of the table.

297: Chapter needs to be expanded and more results to be provided.

308: Figure 5 - I suggest not including figures in Conclusion.

Please prepare the manuscript according to guide for authors. For example, References are not saved correctly.

Author Response

(The authors gave the same response as above.)

Round 2

Reviewer 2 Report

Dear authors

Thank you so much for your positive responses

In my opinion, this manuscript has been significantly improved

It can be accepted in the present form

Regards

Reviewer 3 Report

No comments